# Growth of Renal Cancer Cell Lines Is Strongly Inhibited by Synergistic Activity of Low-Dosed Amygdalin and Sulforaphane

**DOI:** 10.3390/nu16213750

**Published:** 2024-10-31

**Authors:** Sascha D. Markowitsch, Thao Pham, Jochen Rutz, Felix K.-H. Chun, Axel Haferkamp, Igor Tsaur, Eva Juengel, Nathalie Ries, Anita Thomas, Roman A. Blaheta

**Affiliations:** 1Department of Urology and Pediatric Urology, University Medical Center Mainz, 55131 Mainz, Germany; sascha.markowitsch@unimedizin-mainz.de (S.D.M.); jochen.rutz@unimedizin-mainz.de (J.R.); axel.haferkamp@unimedizin-mainz.de (A.H.); igor.tsaur@med.uni-tuebingen.de (I.T.); eva.juengel@unimedizin-mainz.de (E.J.); nathalie.ries@unimedizin-mainz.de (N.R.); anita.thomas@unimedizin-mainz.de (A.T.); 2Department of Urology, Goethe-University, 60590 Frankfurt am Main, Germany; thiphuongthao.pham@unimedizin-ffm.de (T.P.); felix.chun@ukffm.de (F.K.-H.C.)

**Keywords:** amygdalin, sulforaphane, renal cell carcinoma, tumor growth, cell cycling

## Abstract

**Background**: Plant derived isolated compounds or extracts enjoy great popularity among cancer patients, although knowledge about their mode of action is unclear. The present study investigated whether the combination of two herbal drugs, the cyanogenic diglucoside amygdalin and the isothiocyanate sulforaphane (SFN), influences growth and proliferation of renal cell carcinoma (RCC) cell lines. **Methods**: A498, Caki-1, and KTCTL-26 cells were exposed to low-dosed amygdalin (1 or 5 mg/mL), or SFN (5 µM) or to combined SFN-amygdalin. Tumor growth and proliferation were analyzed by MTT, BrdU incorporation, and clone formation assays. Cell cycle phases and cell cycle-regulating proteins were analyzed by flow cytometry and Western blotting, respectively. The effectiveness of the amygdalin–SFN combination was determined using the Bliss independence model. **Results**: 1 mg/mL amygdalin or 5 µM SFN, given separately, did not suppress RCC cell growth, and 5 mg/mL amygdalin only slightly diminished A498 (but not Caki-1 and KTCTL-26) cell growth. However, already 1 mg/mL amygdalin potently inhibited growth of all tumor cell lines when combined with SFN. Accordingly, 1 mg/mL amygdalin suppressed BrdU incorporation only when given together with SFN. Clonogenic growth was also drastically reduced by the drug combination, whereas only minor effects were seen under single drug treatment. Superior efficacy of co-treatment, compared to monodrug exposure, was also seen for cell cycling, with an enhanced G0/G1 and diminished G2/M phase in A498 cells. Cell cycle regulating proteins were altered differently, depending on the applied drug schedule (single versus dual application) and the RCC cell line, excepting phosphorylated Akt which was considerably diminished in all three cell lines with maximum effects induced by the drug combination. The Bliss independence analysis verified synergistic interactions between amygdalin and SFN. **Conclusions**: These results point to synergistic effects of amygdalin and SFN on RCC cell growth and clone formation and Akt might be a relevant target protein. The combined use of low-dosed amygdalin and SFN could, therefore, be beneficial as a complementary option to treat RCC. To evaluate clinical feasibility, the in vitro protocol must be applied to an in vivo model.

## 1. Introduction

The use of complementary and alternative medicine (CAM) has significantly increased among cancer patients in the last decades. A survey on cancer patients in France and Poland indicated that 85% are CAM users [1,2]. The number of cancer patients in Sweden and Germany integrating CAM into their treatment schedules has been estimated to be between 50 and 70% [3,4]. Overall, in Europe, more than 50% of cancer patients utilize CAM [5]. CAM encompasses non-conventional therapies such as phytomedicine and probiotics, and mind-body practices such as acupuncture and hypnosis. Among these options, plant-derived compounds are most frequently used, based on evidence that a balanced diet high in fruits and vegetables reduces cancer risk [6,7].

The integration of plant-derived natural products, either as isolated bioactive compounds or as plant extracts, into conventional cancer treatment is associated with the hope of alleviating disease symptoms, mitigating side effects, boosting the immune system, and preventing disease recurrence. Of the many available herbal drugs, amygdalin and sulforaphane (SFN) are among the ones purported to exert distinct anti-tumor properties.

The natural compound amygdalin, a cyanogenic glycoside found in high concentrations in bitter almonds and fruit kernels such as peaches and apricots, metabolizes into hydrogen cyanide (HCN) in the body through β-glucosidase activity [8]. The proponents of amygdalin therapy assume that HCN selectively accumulates in tumor cells due to their enrichment with β-glucosidase, forcing selective tumor cell destruction [8]. In vitro data point to amygdalin triggered apoptosis induction with an increase of Bax, a decrease of Bcl-2 proteins and alterations of caspase-3 activity, cell growth blockade by acting on the cyclin-cdk axis and akt-mTOR signaling, and invasion suppression via altered integrin α and β receptor expression [9,10,11]. Animal studies show that amygdalin may also down-regulate cancer growth in vivo [12,13]. Nevertheless, its anti-tumor efficacy remains contentious. An old study conducted in 1982, involving 175 cancer patients, did not report a clinical benefit of amygdalin [14]. No high-standard clinical studies have been initiated since then, leaving its anti-tumor potential unresolved. Rather, warnings have been issued that excessive amygdalin consumption may lead to severe cyanide poisoning [15]. During recent years, amygdalin has regained high regard as a cytotoxic and chemopreventive remedy [16,17].

SFN, an isothiocyanate abundant in green vegetables such as broccoli, kale, cabbage and cauliflower, is highly popular among cancer patients, since it is said to prevent oncogenesis and tumor progression [18]. Its anti-cancer effects have been demonstrated on a panel of human tumor cell lines, including bladder, lung, breast, and colon cancer cells, by modulating epigenetic as well as non-epigenetic pathways involving interactions with the Akt/MAPK and the Nrf2/ARE signaling pathways [19]. Detailed information on the molecular mode of action of SFN is provided in [20]. There are further reports pointing to preventive and therapeutic effects of SFN in in vivo tumor models [21,22]. Attenuation of oncogenic pathways along with an increase in tumor suppressor proteins under SFN have been shown in clinical trials [23,24]. SFN has recently been declared one of the “Big Five” phytochemicals targeting cancer stem cells [25].

Renal cell carcinoma (RCC) is the most common type of kidney cancer, accounting for approximately 90% of cases [26]. Once metastasized, RCC is difficult to treat. It does not respond to chemoradiotherapy, and cytokine-based therapy has failed due to insufficient efficacy. The concept of drug targeting with receptor tyrosine kinase inhibitors (TKIs) is also not as efficient as initially thought, due to resistance development [25]. The introduction of immune checkpoint inhibitors (ICIs), often combined with TKIs or other ICIs, has significantly improved the prognosis for patients with metastatic disease. However, this approach is associated with severe immune-related adverse effects [27], and only a minority of patients profit from this type of therapy [28].

Thus, integrating amygdalin and SFN into current treatment could offer potential benefits to RCC patients. The present study was conducted to elucidate growth suppressive properties and the underlying molecular mechanisms of both compounds on RCC cells in vitro, with particular emphasis on low-dosed amygdalin.

## 2. Materials and Methods

### 2.1. Cell Cultures

RCC cell lines Caki-1 and KTCTL-26 were purchased from LGC Promochem (Wesel, Germany). A498 cells were provided by Cell Lines Service (Heidelberg, Germany). All cell lines were grown in RPMI 1640 medium supplemented with 10% fetal calf serum (FCS), 2% HEPES (2-(4-(2-Hydroxyethyl)-1-piperazine)-ethanesulfonic acid) buffer, 1% Glutamax (all Gibco/Invitrogen, Karlsruhe, Germany), and 1% penicillin/streptomycin (both Sigma-Aldrich, München, Germany) at 37 °C in a humidified 5% CO_2_ atmosphere.

### 2.2. Drugs

Amygdalin from apricot kernels (Sigma-Aldrich, Taufkirchen, Germany) was freshly dissolved in cell culture medium at low concentrations of 1 and 5 mg/mL [10,29]. SFN was provided by Biomol, Hamburg, Germany. Based on earlier dose-response analyses, SFN was applied at 5 µM [30,31]. The tumor cells were treated with amygdalin alone, with SFN alone or with both amygdalin and SFN. Controls remained untreated.

### 2.3. Cell Growth and Apoptosis

Cell growth was measured using the 3-(4,5-dimethylthiazol-2-yl)-2,5-diphenyltetrazolium bromide dye reduction test (MTT-assay). Drug treated versus untreated carcinoma cell lines were plated out into 96-multiwell plates (50 µL/well, 1 × 10^5^ cells/mL) and incubated for 24, 48, and 72 h. Then, 10 µL of MTT/well (Roche Diagnostics, Penzberg, Germany) was added for 4 h. Cells were then lysed using sodium dodecyl sulfate (10% SDS in 0.01 M HCl) and incubated overnight at 37 °C. A multi-well ELISA reader determined the absorbance at 570 nm (Tecan Infinite M200, Männedorf, Switzerland). After subtracting background absorbance and matching with a standard curve, the results were expressed as mean cell number. The mean cell number was set at 100% after 24 h of incubation.

Apoptosis was evaluated by the annexin V-FITC Apoptosis Detection kit (BD Pharmingen, Heidelberg, Germany). Tumor cell lines were washed twice with PBS, and then incubated with 5 μL of annexin V-FITC and 5 μL of propidium iodide in the dark for 15 min. Cells were then analyzed using the FACScalibur flow cytometer (BD Biosciences, Heidelberg, Germany). A total of 10,000 events were collected for each sample. CellQuest software version 6.0 (BD Biosciences) served to quantify the percentage of apoptotic (early and late), necrotic, and vital cells.

### 2.4. Bliss Independence Model

Synergistic, additive, or antagonistic effects between SFN and amygdalin were determined using the Bliss independence model [32]. The calculation was based on the MTT data with the assumption that each of the drugs applied, amygdalin and SFN, induces its anti-cancer effect independently by targeting different pathways. The tumor cells were treated with either SFN [5 µM] or amygdalin [1 or 5 µM] or the two combined for 24, 48, and 72 h. The response, in terms of growth inhibition, was normalized to the untreated control at the respective time point (set to 100%). The Bliss predicted inhibition rate IBliss was then calculated by the formula
IBliss = I_amyg_ + I_SFN_ − (I_amyg_ × I_SFN_)

Synergism is defined by IBliss > observed inhibition by the drug combination. IBliss < observed inhibition by the drug combination reflects antagonism, and IBliss = observed inhibition by the drug combination defines an additive effect.

### 2.5. Clonogenic Growth and Cell Proliferation

To evaluate clonogenic tumor growth, 500 cells/well were transferred to 6-well-plates. A 2 mL cell culture medium was additionally added (drug-containing versus drug-free medium). Following 5–10 days of incubation without medium change during this time, cell colonies were fixed and counted. Clones ≥ 50 cells were counted as one colony using a Zeiss ID 03 light microscope (Zeiss AG, Oberkochen, Germany). Tumor cell proliferation was measured using a BrdU (5-bromo-2′-deoxyuridine) cell proliferation enzyme-linked immunosorbent assay (ELISA) kit (Calbiochem/Merck Biosciences, Darmstadt, Germany). Drug treated versus non-treated tumor cells were seeded onto 96-well microtiter plates (50 μL/well, 1 × 10^5^ cells/mL) and incubated with BrdU labeling solution (20 μL/well). After 24 or 48 h, cells were fixed and detected using anti-BrdU mAb, according to the manufacturer’s instructions. Absorbance was measured at 450 nm by a microplate ELISA reader (Tecan Infinite M200, Männedorf, Switzerland). Values were expressed as a percentage compared to untreated controls, set to 100%.

### 2.6. Cell Cycle Analysis

A cell cycle analysis was performed with RCC cell lines grown to subconfluency. To carry out cell cycle analysis, treated and non-treated tumor cell populations were stained with the dye propidium iodide, using the Cycle TEST PLUS DNA Reagent Kit (BD Biosciences, Heidelberg, Germany), and then subjected to flow cytometry with a FACSCalibur (BD Biosciences). A total of 10,000 events per sample were analyzed. Data acquisition was conducted with Cell-Quest software, version 5.1. The cell cycle distribution was calculated using ModFit software, version 3.3 (BD Biosciences). The number of gated cells in the G1, G2/M, or S phase was finally depicted as the percentage of the total number of cells in all phases. Experiments were conducted with both asynchronous and synchronous tumor cells. Tumor cells were synchronized at the G1-S boundary with aphidicolin (1 μg/mL; Sigma-Aldrich) 24 h before starting cell cycle analysis and subsequently resuspended in fresh (aphidicolin-free) medium for 2 h.

### 2.7. Western Blot Analysis

The expression of proteins involved in cell cycle regulation was analyzed in Caki1, A498, and KTCTL-26 cells. The tumor cell lysates were applied to a 7–12% polyacrylamide gel (depending on the protein size to be detected) and electrophoresed for 90 min at 100 V. Proteins were then transferred to nitrocellulose membranes (1 h, 100 V), blocked with nonfat dry milk for 1 h, and incubated overnight with the following monoclonal antibodies: Anti-CDK1/Cdc2 (IgG1, clone 1), anti-pCDK1/Cdc2 (IgG1, clone 44/CDK1/Cdc2 (pY15)), anti-CDK2 (IgG2a, clone 55), anti-Cyclin A (IgG1, clone 25), anti-Cyclin B (IgG1, clone 18), anti-PKBα/AKT (IgG1 clone 55), anti-pAKT (IgG1, Ser472/Ser473, clone 104A282), p27 (IgG1, clone G173-524), p19 (IgG1, clone 52/p19 Skp1; all: BD Pharmingen). Anti-PTEN (clone 26H9) was from Cell Signaling (Leiden, Netherlands, anti-pCDK2 (Thr160) from Thermo Fisher Scientific, Schwerte, Germany, and anti-histone H3 (IgG, clone 3H1), anti-acetylated H3 (aH3; IgG, Lys9, clone C5B11), and anti-histone H4 (IgG, clone L64C1,) were all from Cell Signaling (Leiden, The Netherlands). Anti-acetylated H4 (aH4; Lys8, polyclonal, IgG). Anti-Bax (B-9:sc-7480) and anti-Bcl-2 (N-19:sc-492) were obtained from Santa Cruz (Heidelberg, Germany. HRP-conjugated goat anti-mouse IgG and HRP-conjugated goat anti-rabbit IgG (Upstate Biotechnology, Lake Placid, NY, USA) served as the secondary antibodies. For protein visualization, the membranes were incubated with ECL detection reagent (ECL; Amersham/GE Healthcare, München, Germany). Then, protein bands were analyzed using the Fusion FX7 system (Peqlab, Erlangen, Germany). β-Actin (1:1000; clone AC-15; Sigma-Aldrich) served as the internal control (see also Appendix A). To quantify the intensity of the protein bands, the protein/β-actin intensity ratio was quantified using GIMP 2.8 software.

### 2.8. Statistics

All experiments were carried out three to six times. Statistical significance was calculated using the independent one-way or two-way ANOVA. Differences were considered statistically significant at a *p* value less than 0.05.

## 3. Results

### 3.1. Synergism of Amygdalin–SFN Combination on Tumor Growth

Amygdalin, 1 mg/mL, did not influence tumor growth in all three investigated cell lines, and 5 mg/mL amygdalin only suppressed growth in A498 cells but not in Caki-1 or KTCTL-26 cells (Figure 1). SFN, applied at 5 µM was also only marginally effective. However, when combined with 1 or 5 mg amygdalin the RCC cell number was significantly diminished, compared to untreated controls (Figure 1). Apoptotic events were not induced by either single or dual drug treatment.

Based on the Bliss independence model, synergistic inhibitory effects were already achieved after 24 h in all cell lines in the presence of SFN plus 5 mg/mL amygdalin (Figure 2). Effects on Caki-1 cells were strongest. Following 72 h treatment, synergism was seen even when SFN was combined with the lower amygdalin concentration of 1 mg/mL. Caki-1 and A-498 cells were more sensitive to the drug combination, compared to KTCTL-26 cells.

### 3.2. Suppression of Tumor Proliferation and Clone Development

Tumor proliferation, analyzed by BrdU uptake, was slightly suppressed by 5 mg/mL amygdalin after 48 h (KTCTL-26), whereas SFN diminished the proliferation of all cell lines after 24 h as well as after 48 h (Figure 3). Stronger anti-proliferation effects were induced when the SFN–amygdalin combination was applied. Although 5 mg/mL amygdalin did not alter the cells’ proliferative activity (excepting KTCTL-26, 48 h), it significantly elevated the effects of SFN when the drugs were combined.

Clonogenic growth is shown in Figure 4. Control values were set to 100%, corresponding to 27.0 + 9.2 (A498), 69.0 + 13.7 (Caki-1) or 16.7 + 7.6 (KTCTL-26) clones per well. Amygdalin (5 mg/mL) or SFN significantly reduced the number of tumor clones, independent of the cell line. Combined application was associated with a synergistic effect as calculated by the Bliss independence model.

### 3.3. Influence of Amygdalin and SFN on Cell Cycling

Cell cycle progression was followed on synchronized and non-synchronized cell populations. In the non-synchronized cell model, progression was not congruent. SFN enhanced the number of S-phase cells in all cell lines. The number of G0/G1 Caki-1 and KTCTL-26 cells was reduced, whereas the number of G0/G1 A498 cells remained unchanged. In addition, Caki-1 cells undergoing G2/M were enhanced by SFN, the percentage of A498 in G2/M, however, was lowered (Figure 5, up). Amygdalin (5 mg/mL) slightly elevated G0/G1 phase A498 cells and lowered the number of A498 and KTCTL-26 cells in the S-phase. Simultaneous drug use also altered cell cycle progression. However, superior effects, compared to the monodrug treatment, were only reflected in a G0/G1 increase and loss of G2/M in A498 cells.

In the synchronized cell model, superior effects of the dual drug treatment were apparent in the A498 cells (Figure 5, lower). Differences to the non-synchronized cell model, however, were apparent. SFN reduced the number of S-phase A498 and KTCTL-26 cells but upregulated KTCTL-26 cells in G0/G1. Elevation of G0/G1 phase KTCTL-26 and loss of G2/M phase Caki-1 cells was observed following amygdalin exposure. The amygdalin–SFN combination diminished S-phase A498 and KTCTL-26 cells similarly to amygdalin or SFN alone. In contrast, S-phase Caki-1 cells were significantly elevated under dual treatment.

Cell cycle and apoptosis-regulating proteins were analyzed in non-synchronized cell populations. According to cell cycle progression, SFN and amygdalin did not alter protein expression in all cell lines homogenously (Figure 6, Appendix A, Western blots). Amygdalin reduced cdk1 (A498, Caki-1), pcdk1 (all cell lines), cdk2 (A498, KTCTL-26), pcdk2 (A498), Cyclin A (all cell lines), Cyclin B (A498, Caki-1), pAkt (KTCTL-26), and Bcl-2 (Caki-1). However, it elevated cdk1 (moderately) in KTCTL-26, p19 (all cell lines), and Bax (all cell lines). The tumor suppressor p27 was only upregulated in A498 cells.

SFN diminished only a few proteins: pcdk2 (A498), pAkt (Caki-1, KTCTL-26), and Bcl-2 (KTCTL-26). It, however, enhanced the following proteins: cdk1 (A498, KTCTL-26), pcdk1 (all cell lines), pcdk2 (KTCTL-26), Cyclin A (A498), Cyclin B (KTCTL-26), p19 (all cell lines), p27 (A498), Bax (all cell lines).

Amygdalin or SFN increased the histone, H3, in Caki-1 and KTCTL-26 cells. Acetylated histone, aH3, was increased in A498 (both drugs), Caki-1 (amygdalin only), and KTCTL-26 (amygdalin only). No distinct effects of amygdalin or SFN were exerted on H4, and aH4 was not detectable.

The combined use of both drugs diminished pAkt in all cell lines, much more strongly than each drug alone. An additive effect of the combined drugs was also seen with respect to p27 elevation (A498), Bax elevation and Bcl-2 reduction (both Caki-1) (see also Appendix A).

## 4. Discussion

The MTT assay reflected no growth inhibitory effect on RCC cells when 5 µM SFN was applied alone. This accords with data from Gokay et al. reporting an IC_50_ of 15–20 µM SFN being necessary to significantly block RCC cell growth [33]. Still, the MTT data should not be generalized to the effect that 5 µM SFN exerts no action on RCC cells. BrdU incorporation as well as the clonogenic growth assay documented a significant impact of this dosage on the tumor cells’ proliferative capacity. It is, therefore, conceivable that 5 µM SFN does induce growth suppression. This has been demonstrated by others in a panel of RCC cell lines [30,34].

RCC cell growth was not altered by either 1 or 5 mg/mL amygdalin applied separately, as evidenced by the MTT test. This concurs with an earlier investigation where amygdalin concentrations of 10 mg/mL were required to significantly suppress RCC cell growth in vitro [35]. The efficacy of amygdalin may not only depend on the dosage but on the tumor entity. The IC_50_ value of amygdalin for breast cancer cells has been estimated at approximately 10 mg/mL, 25% growth reduction with 5 mg/mL [36]. Prostate and bladder cancer cell lines were already sensitive to 2.5 mg/mL amygdalin [10,28], whereas a minimum dosage of 30 mg/mL amygdalin was necessary to block pancreatic cancer cell growth [37].

Interpretation of the current data should, therefore, be restricted to the RCC cell model. Here, the SFN-amygdalin drug combination induced a substantial decrease in tumor growth, although monotherapies were without effect. Synergism was apparent even when the lowest amygdalin concentration of 1 mg/mL was used. Clonogenic growth with 5 mg/mL amygdalin was also more effectively inhibited with dual drug application, compared to single drug application. It is of particular interest that the strongest effect of the amygdalin–SFN combination was seen after 5–10 days, indicating that the therapeutic potential of this approach might be maintained for an extended time period.

Low dosing over a longer time period is important since excessive and/or high-dosed consumption of amygdalin can lead to severe cyanide intoxication. In fact, a maximum of 180 mg of HCN can be released from 500 mg of completely hydrolyzed amygdalin [38] and possibly be lethal for humans. A toxic or lethal HCN-concentration following amygdalin consumption has not been established. One patient developed severe metabolic acidosis after ingesting amygdalin with a serum HCN level of 385 µg/dL [39]. However, it has been reported that a patient who had taken twice his usual oral amygdalin dose, and had a blood level of 600 µg/dL cyanide showed no signs of toxicity [40]. Although a critical HCN-serum level following amygdalin metabolization has not been determined, and although our in vitro data cannot be transferred to an in vivo situation, combining amygdalin with SFN allows a considerable reduction in amygdalin dosage, thus limiting the risk of toxicity. Dichotomic effects of amygdalin have recently been reported in a rabbit model: administration of amygdalin at high concentrations was associated with negative effects, whereas low-dosed amygdalin was beneficial [41].

This in vitro study shows that combining SFN with amygdalin exerts a synergistic suppression of tumor cell growth and permits a distinct reduction in amygdalin dosage. How far the amygdalin-SFN combination represents a complementary option to support conventional RCC cancer therapy requires further investigation. Amygdalin has been shown to potentiate the anti-cancer effect of sorafenib on Ehrlich ascites carcinoma in vivo [42] and to act synergistically with cisplatin in a breast cancer cell model [43]. Another investigation has revealed a synergistic effect of an amygdalin-vinblastine mixture on the proliferation of cervical cancer cell lines. Very strong synergism has been observed even at amygdalin concentrations of 1 and 10 µg/mL [44]. Amygdalin has caused potent synergistic effects on osteosarcoma proliferation when combined with chemotherapeutics. Potent synergism between amygdalin and triple drug combinations has also been reported [45].

Sparse data regarding SFN’s influence on RCC cells is available. In our study, 5 µM SFN did not alter tumor cell growth, but did significantly diminish tumor cell clones. Tsaur et al. observed a decrease in RCC cell growth activity, already in the presence of 5 µM SFN [30], whereas others have reported an IC50 for SFN between 6 and 20 µM, depending on the cell line used and drug incubation time [33]. In another study, the treatment of RCC cells with 20 μM SFN was necessary to induce a decrease in cell number [46]. These differences might be due to different RCC cell lines used in the respective studies. Varying SFN preparation techniques may also influence anti-tumor potential. A clinical trial recently investigated the impact of 500 μM (90 mg) SFN given daily to patients with advanced pancreatic cancer. This schedule reduced tumor progression during the first six months after intake. However, some patients suffered from digestive problems, nausea, and emesis [47]. Hence, dual treatment with both SFN and amygdalin may allow dose reduction, limiting unwanted side effects.

How amygdalin and SFN interact is not clear. The Western blot data did not show homogenous additive or synergistic effects in regard to nearly all investigated cell cycle and apoptosis related proteins. Rather, cell signaling was differentially modulated, depending on the cell line and applied drug. It is, therefore, not surprising that cell cycle progression was also altered inconsistently in the presence of amygdalin or SFN alone. A superior effect of the drug combination, compared to mono-drug treatment, was only seen with A498 cells, demonstrating a significant increase of G0/G1 and decrease of G2/M-phase cells in the presence of amygdalin and SFN. In this particular case, cdk1 and 2 along with Cyclin A and B were distinctly blocked under dual drug exposure and might explain the diminished cell growth. Indeed, functional blocking of cdk1 and Cyclin B in A498 cells has been demonstrated to significantly diminish cell growth [34], and Cyclin A expression has been correlated to the G2/M phase in A498 cells [48]. Whether different characteristics of the RCC cell lines used here are responsible for the different cell cycle changes remains unresolved. Caki-1 cells express the wildtype von Hippel-Lindau (VHL) tumor-suppressor protein, and mutations of the *VHL* gene have been detected in KTCTL-26 cells, whereas A498 cells are completely VHL deficient. Therefore, VHL involvement may not explain why synergism was seen in all cell lines in the clonogenic growth assay. Another mode of action may be relevant here. It should be noted that the clones were counted after 5–10 days drug incubation, whereas cell cycling was evaluated 24 h following drug exposure. Therefore, clone data cannot strictly be associated with cell cycle progression.

Only a single protein, pAkt, was more strongly diminished in all cell lines under dual drug treatment, compared to single drug application. Hyperactivation of Akt has been demonstrated in several tumor entities, including RCC, making this protein a relevant target for cancer therapy [49]. RCC resistance development towards TKIs and ICIs involves Akt activation, as recently reported by Aweys et al. [50]. Hence, blocking Akt may optimize treatment and improve the prognosis of RCC patients. The present study shows that the herbal drugs amygdalin and SFN potently suppress Akt, making them valuable tools to fight cancer. In good accordance with this, synergistic action of SFN combined with the natural compound formononetin on cervical cancer cell growth and apoptosis has recently been reported to be triggered by coupled deactivation of Akt [51]. The authors speculated that synergistic blockade of pAkt boosts apoptosis via cdk-cyclin suppression and G0/G1 cell cvcle arrest. We did not investigate apoptosis; however, Bax has been shown to be elevated under SFN/amygdalin co-treatment in all cell lines, and Bcl-2 was distinctly reduced in Caki-1 following combined amygdalin-SFN exposure. Therefore, it cannot be excluded that synergistic deactivation of Akt may also induce apoptotic progression in a synergistic manner. In fact, SFN has already been documented to force apoptosis in pancreatic cancer cells via caspase-3 cleavage [52], and Habib et al. recently documented synergistic effects of the chemotherapeutic agent paclitaxel and SFN on prostate cancer cell apoptosis, triggered by altering Bax and Bcl-2 protein expression and caspase-3 cleavage [53]. Synergistic action on caspase 3 has also been observed when SFN is combined with gemcitabine [54], opening the question on the role of caspase-3 in our cell culture system and treatment protocol.

Respective information on amygdalin is sparse. However, simultaneous application of amygdalin and the anti-diabetic drug metformin induced stronger effects on caspase-3 and apoptosis in hepatocellular carcinoma cells, compared to single drug treatment [55]. This is important, since metformin inhibits Akt as well [56]. Finally, blocking Akt in esophageal squamous cell carcinoma cells by SFN and a further Akt-inhibitor (PP242) synergistically suppressed tumor proliferation and activated apoptotic pathways in vitro. Low-dosed SFN was only moderately effective, whereas the combined application of SFN with PP242 strongly suppressed the growth of tumor xenografts and induced cell apoptosis also in vivo [57].

The exact role of Akt, however, remains unclear. Amygdalin suppressed pAkt in Caki-1 and KTCTL-26 cells when given alone. Therefore, stronger Akt suppression seen in these cells after dual drug treatment might be an additive effect of the two drugs. However, this seems not to be the case with A498 cells, where amygdalin alone had no effect on pAkt expression. This is similar to another publication, where other combined drugs, such as SFN and the tyrosine kinase inhibitor lapatinib, strongly reduced pAkt in gastric cancer cells, although both drugs given separately did not [58]. The authors suggested that SFN may sensitize the tumor cells to lapatinib. Whether a similar scenario holds true for A498 cells is unclear. We assume that the dualistic action of amygdalin and SFN on A498 cells might be (at least partially) initiated via the cdk-Cyclin axis. Indeed, cdk1, cdk2 along with cyclin A and B were all potently diminished by the combination protocol in this cell line.

Further modes of amygdalin-SFN interaction should be considered to explain synergism. Molecular docking studies have shown that amygdalin may interact with broccoli myrosinase which catalyzes the conversion from glucorapharin to SFN [59]. This seems unlikely, however, since pure SFN and not its precursor, glucoraphanin, was used in our study. Jurkowska recently discovered that the isothiocyanate, 4-hydroxybenzyl isothiocyanate, down-regulates the level of mitochondrial rhodanese, forcing increased reactive oxygen species (ROS) production and inhibition of tumor cell growth [60]. This mechanism has not been proven for SFN, although SFN’s impact on ROS has been documented [61]. Since amygdalin suppresses rhodanese as well [8], the synergism of SFN and amygdalin might be traced back to the blockade of rhodanese. It has been suggested that malignant cells have lower levels of the enzyme rhodanese or may even be deficient in rhodanese [8]. Rhodanese functions as an HCN detoxifier by catalysing the formation of thiocyanate from HCN and thiosulfate [8]. Given that rhodanese is lowered or lost in cancer cells, HCN increase could make the cancer cells more vulnerable to other anti-tumor drugs. Christodoulou et al. have documented that amygdalin and cisplatin co-treatment exerts cytotoxic effects on breast cancer but induces chemoprotective effects on normal breast cells [43]. The investigators hypothesized that different expression levels of rhodanese in healthy versus tumor cells might (partially) account for this. This theory may explain why amygdalin has been shown to exert synergism under differing conditions, including drugs with different modes of action.

## 5. Conclusions

Overall, evidence is provided that low-dosed amygdalin and SFN, when applied simultaneously, exert synergistic effects on RCC cell growth in vitro. Embedding both drugs into standard treatment protocols may, therefore, increase tumor response and delay tumor progression. Further attention should be paid to improving amygdalin’s and SFN’s bioavailability and specificity. Nanoencapsulation has been shown to be an innovative technique to enhance drug efficacy. In fact, inclusion complexes of amygdalin with β-cyclodextrin have already been shown to significantly enhance amygdalin’s anti-cancer activity of [62]. Biocompatible and biodegradable alginate-chitosan nanoparticles have also been developed for cancer-specific amygdalin delivery with improved cytotoxic effects and simultaneous protection of healthy cells [63]. A novel investigation points to the synthesis of silk fibroin protein nanoparticles that may allow controlled and long-term SFN release [64]. The in vitro model presented here should now be transferred to an in vivo model to evaluate the clinical feasibility of applying low-dose amygdalin and SFN to treat RCC. It might also be worthwhile to explore the interaction of SFN/amygdalin with immune checkpoint inhibitors (ICI) which have been approved for cancer immunotherapy. In fact, few data published so far are discrepant. Application of SFN to glioblastoma ICI therapy may provide a significant therapeutic effect as stated out by Lee et al. [65]. However, others assume that SFN may impair the anticancer effect of ICI on gastric cancer cells [66]. In this context, SFN may act as a double-edged sword, reducing carcinogenesis but also blocking T cell-mediated immune response [67]. Therefore, ongoing experiments should deal with the relevance of the amygdalin-SFN combination embedded into an ICI treatment protocol in RCC.

## Figures and Tables

**Figure 1 nutrients-16-03750-f001:**
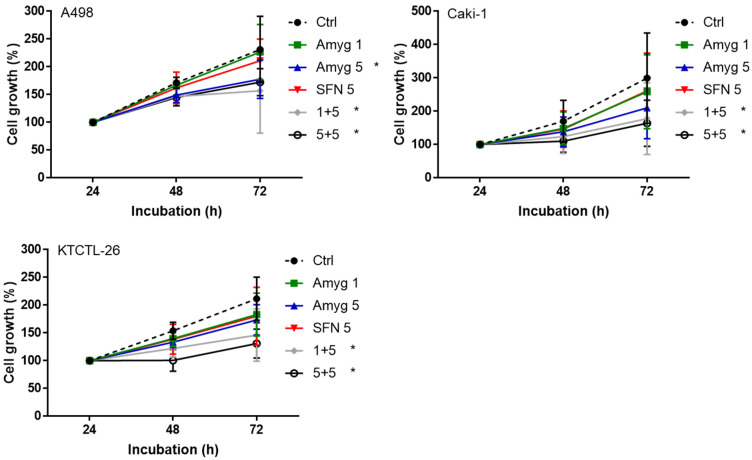
Influence of 5 µM sulforaphane (SFN 5), 1 mg/mL or 5 mg/mL amygdalin (Amyg 1, Amyg 5) or the drug combination (5 µM SFN + 1 mg/mL amygdalin, 1 + 5, 5 µM SFN + 5 mg/mL amygdalin, 5 + 5) on cell growth in Caki1, A498, and KTCTL-26 cells. Cells were exposed to the drugs and evaluated after 24 (set to 100%), 48, and 72 h by the MTT assay. Controls (Ctrl) remained untreated. * indicates significant difference to the controls (One-way ANOVA). *n* = 4.

**Figure 2 nutrients-16-03750-f002:**
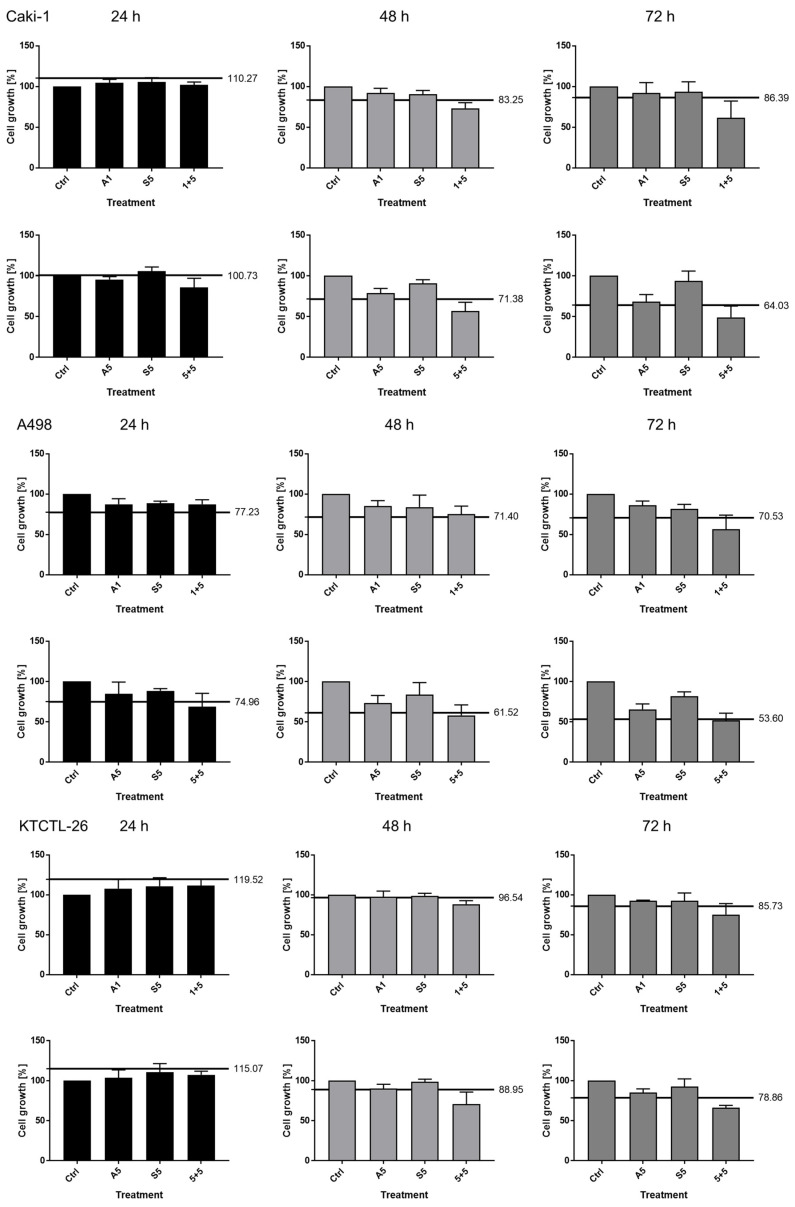
Calculation of synergism (Two-way ANOVA). A498, Caki1 and KTCTL-26 cells were treated with 1 or 5 mg/mL amygdalin (Amyg1, Amyg5) or SFN [5 µM] or with the combination of 1 mg/mL amygdalin + 5 µM SFN (1 + 5) or 5 mg/mL amygdalin + 5 µM SFN (5 + 5) for 24, 48, and 72 h. Growth characteristics were analyzed by the MTT assay and expressed as percentage inhibition, compared to the untreated control (100%). Error bars indicate standard deviation. The calculated effect from the Bliss independence model is displayed as a black line. Effect of the combination above the line = antagonistic effect, on the line = additive effect, below the line = synergistic effect. *n* = 4.

**Figure 3 nutrients-16-03750-f003:**
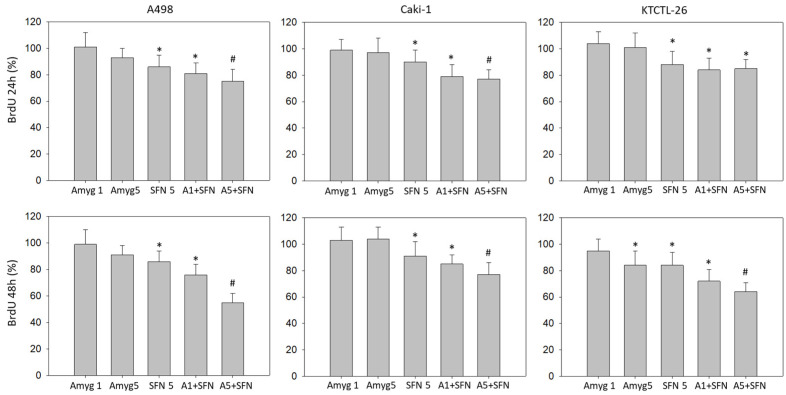
Evaluation of cell proliferation with the BrdU incorporation test. The RCC cell lines A498, Caki-1, and KTCTL-26 were exposed to 1 mg/mL or 5 mg/mL amygdalin (Amyg 1, Amyg 5), to 5 µM sulforaphane (SFN 5) or to the amygdalin-SFN drug combination (A1 + SFN, A5 + SFN) for 24 or 48 h and then subjected to assay. Error bars indicate standard deviation. * indicates significant down-regulation, compared to untreated controls set to 100%. # indicates significant down-regulation of the drug combination, compared to single drug use (Two-way ANOVA). *n* = 3.

**Figure 4 nutrients-16-03750-f004:**
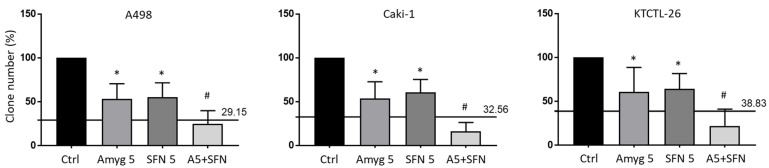
Clonogenic growth assay. Number of A498, Caki-1, and KTCTL-26 cell clones exposed to 5 mg/mL amygdalin (Amyg 5), 5 µM sulforaphane (SFN 5), or both (A5 + SFN). Controls (Ctrl) were incubated in cell culture medium alone. Error bars indicate standard deviation. * indicates significant difference to untreated controls, set to 100%. # indicates significant difference of the drug combination, compared to single drug use (Two-way ANOVA). The calculated effect from the Bliss independence model is displayed as a black line. Effect of the combination above the line = antagonistic effect, on the line = additive effect, below the line = synergistic effect. *n* = 3.

**Figure 5 nutrients-16-03750-f005:**
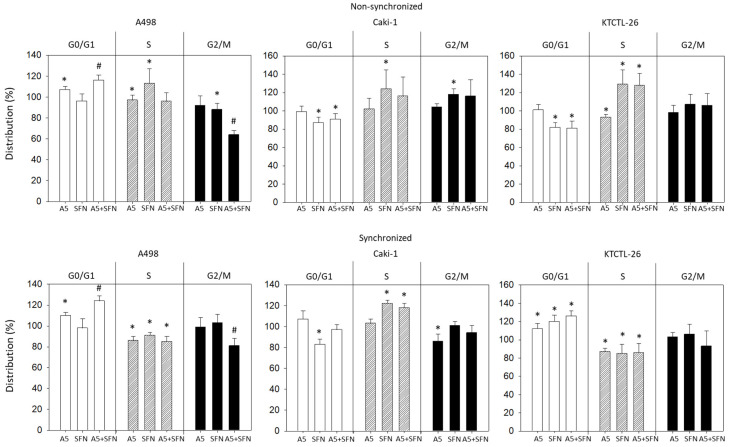
Influence of 5 mg/mL amygdalin (A5), 5 µM sulforaphane (SFN), or both (A5 + SFN) on proportionate G0/G1, S, and G2/M-phases in A498, Caki-1, and KTCTL-26 cell lines. The upper part of the figure shows data from non-synchronized cells, the lower part from synchronized cells. Mean of three experiments. * indicates significant difference to untreated controls (set to 100%). # indicates significant difference of the drug combination, compared to single drug use (Two-way ANOVA).

**Figure 6 nutrients-16-03750-f006:**
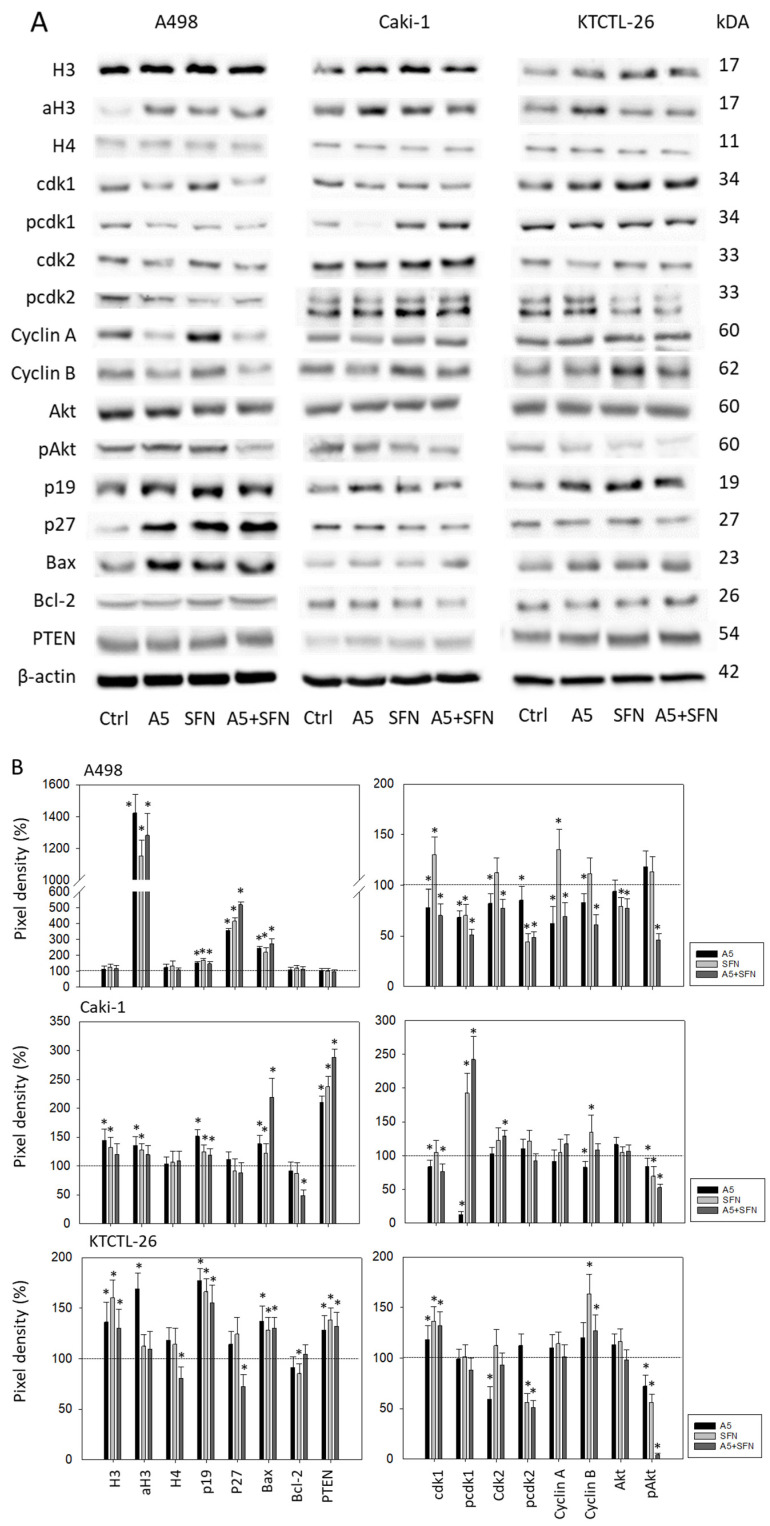
(**A**): Western blot of cell cycle and apoptosis-related proteins from A498, Caki-1, and KTCTL-26 cell lysates. Tumor cells were pretreated with 5 mg/mL amygdalin (A5), 5 µM sulforaphane (SFN), or both compounds (A5 + SFN) for 24 h. Controls (Ctrl) remained untreated. β-actin served as the internal control. One representative from three separate experiments is shown. (**B**): Pixel density. Values are depicted as a percentage, related to the untreated control cells (set to 100% and indicated by a horizontal line). * indicates significant difference to controls.

## Data Availability

Data are contained within the article.

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
