# Peer review of "Growth of Renal Cancer Cell Lines Is Strongly Inhibited by Synergistic Activity of Low-Dosed Amygdalin and Sulforaphane"

_nutrients, 2024, doi:10.3390/nu16213750_

Round 1

Reviewer 1 Report

Comments and Suggestions for Authors

Comments for manuscript ID nutrients-3240494

This study was aimed to exploring the role of several plant extracts on renal cells, and it found that the synergistic activity of amygdalin and sulforaphane (SFN) inhibits the cell activity, cell cycling, and the phosphorylation of Akt.

Major Comments:

1.        The section of abstract must be re-edited, and especially the description of results.

2.        In the section of introduction, A survey… in line 39, why choice two citations in #1, 2? Importantly, more information of amygdalin and sulforaphane (SFN) on anti-tumor effects should added in this section, including the regulatory mechanism and/or functional targets.

3.        In materials and methods, the item number or reagent number must be added.

4.        The blot bands should be quantitively analyzed. What are phosphorylated protein sites of p-Akt? It should be labeled and analyzed in the section of discussion.

5.        In the file of original images, how to distinguish which position is the target protein? Why isn't there a band with protein marker? Not even labeled for grouping?

6.        What is the point of synergy between amygdalin and SFN? Although there is some data, how can it be explained?

Comments on the Quality of English Language

Extensive editing of English language required

Author Response

Answers to the comments of referee 1

Comment 1: The section of abstract must be re-edited, and especially the description of results.

Our answer. The abstract section has been modified (particularly results) and reads now:

Abstract: Background: Plant derived isolated compounds or extracts enjoy great popularity among cancer patients, although knowledge about their mode of action is unclear. The present study investigated whether the combination of two herbal drugs, the cyanogenic diglucoside amygdalin and the isothiocyanate sulforaphane (SFN), influences growth and proliferation of renal cell carcinoma (RCC) cell lines. Methods: A498, Caki-1, and KTCTL-26 cells were exposed to low-dosed amygdalin (1 or 5 mg/ml), or SFN (5 µM) or to combined SFN-amygdalin. Tumor growth and proliferation were analyzed by MTT, BrdU incorporation, and clone formation assays. Cell cycle phases and cell cycle-regulating proteins were analyzed by flow cytometry and West-ern blotting, respectively. The effectiveness of the amygdalin-SFN-combination was determined using the Bliss independence model. Results: 1 mg/ml amygdalin or 5 µM SFN, given separately, did not suppress RCC cell growth, and 5 mg/ml amygdalin only slightly diminished A498 (but not Caki-1 and KTCTL-26) cell growth. However, already 1 mg/ml amygdalin potently inhibited growth of all tumor cell lines when combined with SFN. Accordingly, 1 mg/ml amygdalin suppressed BrdU incorporation only when given together with SFN. Clonogenic growth was also drastically reduced by the drug combination, whereas only minor effects were seen under single drug treatment. Superior efficacy of co-treatment, compared to monodrug exposure, was also seen for cell cycling, with an enhanced G0/G1 and diminished G2/M phase in A498 cells. Cell cycle regulating proteins were altered differently, depending on the applied drug schedule (single versus dual application) and the RCC cell line, excepting phosphorylated Akt which was considerably diminished in all three cell lines with maximum effects induced by the drug combination. The Bliss independence analysis verified synergistic interactions between amygdalin and SFN. Conclusions: These results point to synergistic effects of amygdalin and SFN on RCC cell growth and clone formation and Akt might be a relevant target protein. The combined use of low-dosed amygdalin and SFN could, therefore, be beneficial as a complementary option to treat RCC. To evaluate clinical feasibility, the in vitro protocol must be applied to an in vivo model.

Comment 2: In the section of introduction, A survey… in line 39, why choice two citations in #1, 2? Importantly, more information of amygdalin and sulforaphane (SFN) on anti-tumor effects should added in this section, including the regulatory mechanism and/or functional targets.

Our answer: The introduction section starts with two different surveys, one conducted in France, one conducted in Poland. It was, therefore, necessary to correctly cite these studies with reference 1 to be related to France, and reference 2 to be related to the Polish study. Concerning the molecular mode of action of both compounds, we have now included more information and respective references. Still, we kept the informations short, not to overload the manuscript. The respective introduction parts read now (lines 62-66): “In vitro data point to amygdalin triggered apoptosis induction with an increase of Bax, a decrease of Bcl-2 proteins and alterations of caspase-3 activity, cell growth blockade by acting on the cyclin-cdk axis and akt-mTOR signaling, and invasion suppression via altered integrin α and β receptor expression [9-11]”.

“SFN, an isothiocyanate abundant in green vegetables such as broccoli, kale, cabbage and cauliflower, is highly popular among cancer patients, since it is said to prevent oncogenesis and tumor progression [18]. Its anti-cancer effects have been demonstrated on a panel of human tumor cell lines, including bladder, lung, breast, and colon cancer cells, by modulating epigenetic as well as non-epigenetic pathways, involving interactions with the Akt/MAPK and the Nrf2/ARE signaling pathways [19]. Detailed information on the molecular mode of action of SFN are provided in [20].

Comment 3: In materials and methods, the item number or reagent number must be added.

Our answer: Vendor/Catalog No and RRID codes are now listed in supplements (S1).

Antibody                                            Vendor/Catalog No                            RRID

Anti CDK1(clone 1)   :                       BD Pharmingen 610037                    AB_397454

anti-pCDK1/Cdc2 (clone 44)            BD Pharmingen 612306                    AB_399621

anti-CDK2 (IgG2a, clone 55)            BD Pharmingen 610145                    AB_397547

anti-Cyclin A (IgG1, clone 25)          BD Pharmingen 611268                    AB_398797

anti-Cyclin B (IgG1, clone 18)          BD Pharmingen 610220                    AB_397617

anti-PKBα/AKT (IgG1 clone 55)      BD Pharmingen 610861                    AB_398180

anti-pAKT (clone 104A282)             BD Pharmingen 550747                    AB_393864

anti-p19 (clone 52/p19 Skp1)             BD Pharmingen 610530                    AB_397887

Anti-PTEN (Clone 26H9)                  Cell Signaling 9556                           AB_331153

anti-pCDK2 (Thr160)                        Thermo Fisher PA5-104849              AB_2816322

anti-histone H3 (clone 3H1)              Cell Signaling 9717                           AB_331222

anti-acetylated H3 (clone C5B11)     Cell Signaling 9649                           AB_823528

anti-histone H4 (clone L64C1)          Cell Signaling 2960                           AB_1147657

anti-acetylated H4 (polyclonal)          Millipore 07-328                                AB_310524

Anti-Bax (B-9:sc-7480)                     Santa Cruz sc-7480                            AB_626729

anti-Bcl-2 (N-19:sc-492)                    Santa Cruz sc-492                              AB_2064290

anti-β-Actin (clone AC-15)               Sigma-Aldrich A1978                       AB_476692

Comment 4: The blot bands should be quantitatively analyzed. What are phosphorylated protein sites of p-Akt? It should be labeled and analyzed in the section of discussion.

Our answer: We did pixel analysis which is now depicted in figure 6B. Materials and Methods, chapter “Western Blot Analysis”, reads now: “To quantify the intensity of the protein bands, the protein/β-actin intensity ratio was quantified using GIMP 2.8 software”. Phosphorylated sites of pAkt have already been given in the methods part (Ser472/Ser473).

Comment 5: In the file of original images, how to distinguish which position is the target protein? Why isn't there a band with protein marker? Not even labeled for grouping?

Our answer: In fact, all western blots were paralleled by a protein ladder to identify the correct protein bands. The supplemental figures (S2) have now been improved in this matter.

Comment 6: What is the point of synergy between amygdalin and SFN? Although there is some data, how can it be explained?

Our answer: We have no final explanation on the exact mechanism contributing to synergistic effects. Still, we have deepened this point in the discussion section, which now reads (lines 435-451):

“The present study shows that the herbal drugs amygdalin and SFN potently suppress Akt, making them valuable tools to fight cancer. In good accordance, synergistic action of SFN combined with the natural compound formononetin on cervical cancer cell growth and apoptosis has recently been reported to be triggered by coupled deactivation of Akt [51]. The authors speculated that synergistic blockade of pAkt boosts apoptosis via cdk-cyclin suppression and G0/G1 cell cvcle arrest. We did not investigate apoptosis, however, Bax has been shown to be elevated under SFN/amygdalin co-treatment in all cell lines, and Bcl-2 was distinctly reduced in Caki-1 following combined amygdalin-SFN exposure. Therefore, it cannot be excluded that synergistic deactivation of Akt may also induce apoptotic progression in a synergistic manner. In fact, SFN has already been documented to force apoptosis in pancreatic cancer cells via caspase-3 cleavage [52], and Habib et al. recently documented synergistic effects of the chemotherapeutic agent paclitaxel and SFN on prostate cancer cell apoptosis, triggered by altering Bax and Bcl-2 protein expression and caspase-3 cleavage [53]. Synergistic action on caspase 3 has also been observed when SFN has been combined with gemcitabine [54], opening the question on the role of caspase-3 in our cell culture system and treatment protocol.

Respective information on amygdalin are sparse. However, simultaneous application of amygdalin and the anti-diabetic drug metformin induced stronger effects on caspase-3 and apoptosis in hepatocellular carcinoma cells, compared to single drug treatment [55]. This is important, since metformin inhibits Akt as well [56]. Finally, blocking Akt in esophageal squamous cell carcinoma cells by SFN and a further Akt-inhibitor (PP242) synergistically suppressed tumor proliferation and activated apoptotic pathways in vitro”.

Lines 463-468: “The authors suggested that SFN may sensitize the tumor cells to lapatinib. Whether a similar scenario holds true for A498 cells is unclear. We assume that the dualistic action of amygdalin and SFN on A498 cells might be (at least partially) initiated via the cdk-Cyclin axis. Indeed, cdk1, cdk2 along with cyclin A and B were all potently diminished by the combination protocol in this cell line”.

Reviewer 2 Report

Comments and Suggestions for Authors

The number of renal cell carcinoma patients are increasing worldwide. At clinically advanced stages, the current therapy is not effective enough. In this manuscript, Markowitsch SD et al demonstrate that amygdalin (A) and sulforaphane (SFN) are effective to suppress three renal cancer cell lines. This manuscript is well-organized; however, following points should be clarified.

Major points

#1: In figure 6, acetyl-H4(aH4) may not need to show the row.

#2: In figure 6, Bax was elevated after A5+SFN treatment in all cell lines. Did A5+SFN activate apoptotic signaling, such as caspase-3?

Minor points

##1: The lines are overlapped and hard to see in Figure 1.

Author Response

Answers to the comments of referee 2

Comment 1: In figure 6, acetyl-H4(aH4) may not need to show the row.

Our answer: We have deleted the row aH4.

Comment 2: In figure 6, Bax was elevated after A5+SFN treatment in all cell lines. Did A5+SFN activate apoptotic signaling, such as caspase-3?

Our answer: Indeed, we assume that Bax may be involved in amygdalin/SFN-triggered apoptosis. We did not investigate this issue, and no data are available for RCC in the context of amygdalin and/or SFN treatment. Still, we have deepened this point in the discussion section, which now reads (lines 435-454):

“The present study shows that the herbal drugs amygdalin and SFN potently suppress Akt, making them valuable tools to fight cancer. In good accordance, synergistic action of SFN combined with the natural compound formononetin on cervical cancer cell growth and apoptosis has recently been reported to be triggered by coupled deactivation of Akt [51]. The authors speculated that synergistic blockade of pAkt boosts apoptosis via cdk-cyclin suppression and G0/G1 cell cvcle arrest. We did not investigate apoptosis, however, Bax has been shown to be elevated under SFN/amygdalin co-treatment in all cell lines, and Bcl-2 was distinctly reduced in Caki-1 following combined amygdalin-SFN exposure. Therefore, it cannot be excluded that synergistic deactivation of Akt may also evoke apoptotic progression in a synergistic manner. In fact, SFN has already been documented to force apoptosis in pancreatic cancer cells via caspase-3 cleavage [52], and Habib et al. recently documented synergistic effects of the chemotherapeutic agent paclitaxel and SFN on prostate cancer cell apoptosis, triggered by altering Bax and Bcl-2 protein expression and caspase-3 cleavage [53]. Synergistic action on caspase 3 has also been observed when SFN has been combined with gemcitabine [54], opening the question on the role of caspase-3 in our cell culture system and treatment protocol.

Respective information on amygdalin are sparse. However, simultaneous application of amygdalin and the anti-diabetic drug metformin induced stronger effects on caspase-3 and apoptosis in hepatocellular carcinoma cells, compared to single drug treatment [55]. This is important, since metformin inhibits Akt as well [56]. Finally, blocking Akt in esophageal squamous cell carcinoma cells by SFN and a further Akt-inhibitor (PP242) synergistically suppressed tumor proliferation and activated apoptotic pathways in vitro”.

Minor points: The lines are overlapped and hard to see in Figure 1.

Our answer: Unfortunately, MTT-values for single drug treatment are quite similar making it difficult to discriminate between the treatments. Still, the data for dual treatment are depicted clearer. We have now color-coded the lines for better visibility.

Reviewer 3 Report

Comments and Suggestions for Authors

This is a study attempting to prove that amygdalin and sulforaphane, although inefficient when acting separately, are apt to inhibit cancer cells growth when given together.

It is not clear where the authors used t-test and where Wilcoxon–Mann–Whitney U test. The authors say nothing about data distribution.

For the computations underlying figures 2-5, ANOVA or Kruskal–Wallis test seem to be the adequate statistical tests.

Many statistical computations have been performed - it seems that the authors have omitted to employ a strategy for dealing with the multiple comparisons problem, in the absence of which it is doubtful whether the results maintain their statistical significance.

Let us admit that the results are statistically significant - are they clinically significant? Does a 10-40 percent inhibition of cell growth translate into clinical results? How does this compare with the growth inhibition induced by conventional cytotoxics?

Author Response

Answers to the comments of referee 3

Comment 1: It is not clear where the authors used t-test and where Wilcoxon–Mann–Whitney U test. The authors say nothing about data distribution. For the computations underlying figures 2-5, ANOVA or Kruskal–Wallis test seem to be the adequate statistical tests. Many statistical computations have been performed - it seems that the authors have omitted to employ a strategy for dealing with the multiple comparisons problem, in the absence of which it is doubtful whether the results maintain their statistical significance.

Our answer: We are thankful for this comment. Wilcoxon–Mann–Whitney U test has not been done in the present investigation. This note has erroneously been transferred from another data set where a non-parametric test has been used. We apologize for this mistake. In fact, we have performed a two-way ANOVA (Fig.1) for the growth analyses and a one-way ANOVA for the remaining experiments (Fig. 3-5). We have updated the statistics section accordingly which reads now: “All experiments were carried out three to six times. Statistical significance was calculated using the independent one-way or two-way ANOVA. Differences were consid-ered statistically significant at a p value less than 0.05”. Legend of fig. 1 reads: “Controls (Ctrl) remained untreated. * indicates significant difference to the controls. One-way ANOVA. n = 4”. Legend of figure 2 reads: “Calculation of synergism (Two-way ANOVA)”. Legend of figure 3 reads. “# indicates significant down-regulation of the drug-combination, compared to single drug use. Two-way ANOVA. n = 3”. Statistical information are now also given in the legends of fig. 4 and 5 (Two-way ANOVA).  

Comment 2: Let us admit that the results are statistically significant - are they clinically significant? Does a 10-40 percent inhibition of cell growth translate into clinical results? How does this compare with the growth inhibition induced by conventional cytotoxics?

Our answer: We absolutely agree with the referee that the in vitro data presented here may not be transferred to the in vivo situation. This has been stated out in the final para of the discussion section which reads: “The in vitro model presented here should now be transferred to an in vivo model to evaluate the clinical feasibility of applying low-dose amygdalin and SFN to treat renal cell cancer”. We also would like to emphasize that we do not argue that SFN-amygdalin may replace conventional treatment. Rather the opposite is the case. We point out in “Discussion”: “Embedding both drugs into standard treatment protocols may, therefore, increase tumor response and delay tumor progression”. Introduction reads: “Thus, integrating amygdalin and SFN into current treatment could offer potential benefits to RCC patients”.

Our study has been done to investigate whether low-dosed amygdalin combined with SFN may induce anti-tumor effects in an in vitro RCC model. It was not our intention to compare this protocol with the standard RCC treatment. However, it might be worthwhile to do this, particularly exploring the interaction of SFN/amygdalin with immune checkpoint inhibitors (ICI) which have been approved for cancer immunotherapy. In fact, few data published so far are discrepant. Application of SFN to glioblastoma ICI therapy may provide a significant therapeutic effect as stated out by Lee et al. (Lee et al. Phytochemicals in Cancer Immune Checkpoint Inhibitor Therapy. Biomolecules. 2021;11:1107.). However, others assume that SFN may impair the anticancer effect of ICI on gastric cancer cells (Zhang et al. Sulforaphane impaired immune checkpoint blockade therapy through activating ΔNP63α/PD-L1 axis in gastric cancer. Mol Carcinog. 2024;63:1611-1620). In this context, SFN may act as a double-edged sword, reducing carcinogenesis but also blocking T cell-mediated immune response (Liang et al. Sulforaphane as anticancer agent: A double-edged sword? Tricky balance between effects on tumor cells and immune cells. Adv Biol Regul. 2019;71:79-87). Therefore, ongoing experiments should deal with the relevance of the amygdalin-SFN combination embedded into an ICI treatment protocol in RCC. We have now added in the discussion section (lines 502-510): “It might also be worthwhile to explore the interaction of SFN/amygdalin with immune checkpoint inhibitors (ICI) which have been approved for cancer immunotherapy. In fact, few data published so far are discrepant. Application of SFN to glioblastoma ICI therapy may provide a significant therapeutic effect as stated out by Lee et al. [65]. However, others assume that SFN may impair the anticancer effect of ICI on gastric cancer cells [66]. In this context, SFN may act as a double-edged sword, reducing carcinogenesis but also blocking T cell-mediated immune response [67]. Therefore, ongoing experiments should deal with the relevance of the amygdalin-SFN combination embedded into an ICI treatment protocol in RCC”.
